# Tin/Tin Oxide Nanostructures: Formation, Application, and Atomic and Electronic Structure Peculiarities

**DOI:** 10.3390/nano13172391

**Published:** 2023-08-22

**Authors:** Poting Liu, Vladimir Sivakov

**Affiliations:** 1Department Functional Interfaces, Leibniz Institute of Photonic Technology, Albert-Einstein Str. 9, 07745 Jena, Germany; poting.liu@leibniz-ipht.de; 2Institute of Physical Chemistry, Friedrich Schiller University Jena, Helmholtzweg 4, 07743 Jena, Germany

**Keywords:** tin, tin oxides, nanostructures, thin films, synchrotron, X-ray spectroscopy, batteries, energy conversation and storage, gas sensor, surface-enhanced Raman scattering

## Abstract

For a very long period, tin was considered one of the most important metals for humans due to its easy access in nature and abundance of sources. In the past, tin was mainly used to make various utensils and weapons. Today, nanostructured tin and especially its oxide materials have been found to possess many characteristic physical and chemical properties that allow their use as functional materials in various fields such as energy storage, photocatalytic process, gas sensors, and solar cells. This review discusses current methods for the synthesis of Sn/SnO_2_ composite materials in form of powder or thin film, as well as the application of the most advanced characterization tools based on large-scale synchrotron radiation facilities to study their chemical composition and electronic features. In addition, the applications of Sn/SnO_2_ composites in various fields are presented in detail.

## 1. Introduction

Tin, one of the earliest metals known by human beings in 3500 BC, is still one of the most important metals today. In the very ancient period, soft copper was combined with a relative amount of tin to make bronze materials, which are much harder and can be applied in various tools and weapons with high strength and long lifetime. During recent centuries, with the development of fundamental scientific studies, a deeper understanding of tin’s chemical and physical properties has been constructed. Tin and its compounds such as tin oxides and tin sulfides, have been widely used as functional materials in electronics, chemical engineering, energy storage, and bio-photonics [1].

Among the wide range of tin-related materials, metallic tin and its oxides, which are the most usually existing tin states in nature, particularly attracting attention from scientists because of their unique structures and properties [2]. Metallic tin is a semimetal with atomic number 50 in the periodic table of elements. In general, there are three known tin oxide forms: SnO, SnO_2_, and Sn_3_O_4_. SnO is a p-type semiconductor with a dielectric constant of 15 and a bandgap about 2.4–2.7 eV [3]. At high temperature above 400 °C, SnO is thermodynamically unstable and disproportionate to metallic tin (0) and tin (IV) oxide. SnO_2_ is the most abundant and thermodynamically stable tin state in the nature. In ambient conditions, the crystal of SnO_2_ typically shows a tetragonal rutile structure. Stoichiometric SnO_2_ is generally be considered as an n-type semiconductor with a wide bandgap about 3.6 eV [4]. Sn_3_O_4_ is a metastable intermediate between SnO_2_ and SnO. It has been discussed that Sn_3_O_4_ has a layered structure which is held together by van der Waals forces [5]. As Sn_3_O_4_ has both Sn(IV) and Sn(II) ions, its electronic structure contains features form both SnO_2_ and SnO. In the reported literature, it is expected to be a p-type semiconductor with a bandgap in the range of 2.2–3.0 eV [5].

During the last couple of decades, nanoscale materials and nanostructures have attracted substantial attention due to the quantum size effect-induced novel properties when the size of a material is comparable to the Bohr radius [6,7]. Among all metal oxide semiconductors, nanomaterials of tin/tin oxides have received considerable notification and have found potential applications in various areas such as solar cells [8], lithium batteries [9], gas sensors [10], and catalysis [11,12], as shown in Figure 1. At the nanoscale, the optoelectronic properties of SnO_2_ such as optical bandgap, conductivity, and photoluminescence can be controlled via the incorporation of impurity or defects [13]. Moreover, Sn/SnO_2_ materials have high chemical and mechanical stability, have high abundance in nature, and are environmentally friendly. All these properties and advantages make tin/tin oxides promising candidates for constructing various next-generation energy storage and optoelectronic devices. In this review, various synthesis routes for fabricating nanostructured tin/tin oxides are presented, together with the introduction of state-of-the-art surface sensitive methods using large-scale synchrotron radiation characterization tools to study their unique atomic and electronic structure. Moreover, the different application potential for these functional materials is discussed. It is expected to provide an overview for the development of tin/tin oxide nanomaterials in recent years and promote related research.

## 2. Fabrication of Tin-Based Nanostructures and Thin Films

To further promote the widespread use of Sn/SnO_2_ composite (nano)powders and thin films, various fabrication methods have been developed with respect to the required properties, structures, and large-scale production. Among all fabrication methods, solid-state based, solution-based, and vapor-based methods can be used to classify the different routes.

### 2.1. Solid-State Methods

In solid-state methods, reactions between solid reactants are usually generated with additional heat or mechanic treatment. These methods are usually simple and efficient, with one synthesis step, whereby many materials can be obtained with high concentration. Furthermore, by adjusting different parameters, it is possible to control the size of the obtained materials.

Sinha et al. fabricated spherical Sn/SnO_2_ nanoparticles with 50 nm diameter on a gram level using a focused solar irradiation approach [14]. By stirring a strong NaOH solution with SnCl_2_ power, black microplates of SnO were observed. Upon further solar irradiation, a transformation into high-bandgap Sn/SnO_2_ nanoparticles was obtained. On the other hand, solid-state methods are mainly applied to synthesize pure tin oxide materials, such as SnO or SnO_2_. For example, Li et al. synthesized tin oxide nanoparticles in the size range of 3–15 nm through a convenient inexpensive and efficient one-step solid-state process [15]. Tin chloride, KCl, and KOH are typical reactants which can be mixed, accompanied by the emission of water vapor, followed by a wash and annealing step. According to the annealing temperature controlling, SnO_2_ nanoparticles with different sizes can be synthesized. The obtained SnO_2_ nanoparticles have high yield and gas sensitivity to EtOH, H_2_, and CO. Yang et al. applied a mechanochemical reaction between SnCl_2_ and Na_2_CO_3_ with NaCl as a diluent, followed by heat treatment at 600 °C to make tin oxide nanocrystals with an average crystal size of about 28 nm [16]. Chakravarty et al. also applied this method to synthesis mesoporous tin oxide in the range of 6–12 nm with a large surface area of 265 ± 16 m^2^ g^−1^. The so-obtained mesoporous tin oxide can be used as an advanced sorbent material for biological applications [17]. Apart from traditional solid-based methods to get 3D nanoparticles, more efforts are currently going toward fabricating 2D materials because of their unique physical and chemical properties. Jiang et al. obtained a 2D SnO nanosheet via mechanical exfoliation using Scotch tape, as shown in Figure 2 [18]. By adjusting the peeling process, nanolayers with varying thicknesses were formed, and the authors further found that the obtained nanostructures showed physical properties that were strongly thickness-dependent. The bandgap could be tuned from the IR range (0.60 eV) for bulk SnO to the UV range (3.65 eV) for the monolayer. On the other hand, it was reported that 2D SnO could be a good precursor for the synthesis of Sn/SnO_2_ composites, as presented in the next section related to wet chemical Sn/SnO_2_ composite formation. Additionally, it should be pointed out that classical solid-state methods such as mechanical milling can be used for the formation of Sn/SnO_2_ composites where tin oxide and metallic tin are mixed and milled in different stoichiometric ratios. Sivashanmugam et al. showed that a matrix in which metallic tin can be distributed without aggregation is essential for realizing Sn/SnO_2_ anodes with high cyclability [19].

### 2.2. Solution-Based Methods

Another popular route to synthesize Sn-based materials is based on the solution process, which usually includes an intensive chemical reaction in solution. The materials obtained using these methods are more uniformly distributed, and the experimental conditions are easily controlled.

One of the most important applications of Sn/SnO_2_ nanostructures is in electrodes for lithium-ion batteries, sensors, and supercapacitors. To increase the capacity, it is necessary to combine them with carbon materials, and solution processes provide an easy way to realize the synthesis. For example, Zhu et al. created a roughly 3 µm flower-like Sn/SnO_2_ graded-structure with excellent ethanol gas-sensing properties using 2D SnO sheets and a hydrothermal technique [20]. Hassan et al. applied an easy solution process using SnCl_2_·2H_2_O as a precursor and CMK-3 as a carbon framework [21]. After stirring, drying, and annealing processes, uniformly Sn/SnO_2_ embedded within the carbon pore walls could be obtained, exhibiting high and stable performance in lithium-ion batteries. Using a similar process, Wang et al. obtained evenly distributed Sn/SnO_2_ nanoparticles with 5 nm average diameter in multilayers of graphene sheets, presenting a hollow spherical structure [22].

Another technique to synthesize Sn/SnO_2_ nanoparticles is the electrochemical approach [23]. Saito et al. developed a surfactant-free direct-current electrolysis method using KCl as the electrolyte, as shown in Figure 3a–c [24]. By controlling the applied voltage and the concentration of the electrolyte, the obtained Sn/SnO_2_ particle size could be tuned from 200 to 1000 nm. Santiago-Giraldo et al. studied the reaction time when mixing SnCl_2_, cetyltrimethylammonium bromide, and ammonium hydroxide directly in an aqueous medium under reflux [25]. More photocatalytic activity was found with a longer reaction time.

In addition to the abovementioned methods, other solution-based ways include the hydrothermal process and sol–gel method. The hydrothermal method is one of the most classic routes to synthesize nanoparticles, and it gives a flexible way to control conditions. By applying high-pressure conditions, intensive chemical reactions can happen in a solution above its boiling point. Using this method, it is easier to fabricate tin oxide materials. For example, by utilizing the redox activity of dicarboxylic acid, it is possible to realize significant control over the composition and morphology of the synthesized SnO_2_ structures, as reported by Zima et al. and shown in Figure 3d–g [26]. The control over dicarboxylic acid can lead to the formation of single-phase Sn_3_O_4_ in hexagonal nanoplates and SnO_2_/Sn_3_O_4_ mixed phases in hierarchical flower-like structures. Furthermore, Akhir et al. found it is easy to change the crystallite size of SnO_2_ nanostructures by simply varying the precursor concentration, as well as reaction temperature and duration, during the hydrothermal synthesis [27]. More specifically, the crystal size could be obtained in the range of 7.88 to 18.41 nm.

### 2.3. Vapor-Based Methods

Vapor-state methods mainly applied for thin-film deposition. These processes generally include the evaporation of the primary precursors, followed by vigorous reactions with or without co-reactants. The benefits of these methods are well-controlled deposition conditions, uniformly distributed layers, high purity, and reproducibility. However, vapor-based methods usually require high-vacuum equipment and, therefore, bring high cost to large-scale production. Nevertheless, the electronic industry widely employs these technologies to obtain high-quality functional layers.

Sputtering is a promising technique as it can bring less contamination and high purity [28,29]. Sn/SnO_2_ can be used as the Sn and O sources in the sputtering technique. For example, Hsu et al. reported that, by controlling the sputtering conditions such as temperatures and pressure when using the robust Sn/SnO_2_ mixed target, the deposited films could be tuned from pure n-type SnO_2_ to pure p-type SnO, with a p-type Hall mobility of up to 2 cm^2^ V^−1^ s^−1^. One typical morphology of SnO_2_ obtained by sputtering deposition is shown in Figure 4c [30,31,32]. On the other hand, the same technique can be used for the formation of Sn/SnO_2_ composites, as Mohamed et al. reported the feasibility of tuning the composition Sn/SnO_2_ during the deposition by controlling the sputtering conditions [33].

Chemical vapor deposition (CVD) has been widely used in the commercial deposition industry because of its inexpensive nature and flexibility in the production of synthesis thin films and 1D nanostructures with high quality [38,39,40,41]. Many studies have described the CVD process of Sn-based thin films using SnCl_4_ and SnI_4_ as the tin precursors. In our previous studies, we also presented the deposition of Sn/SnO_2_ thin films using tin(IV) tert-butoxide as the tin and oxygen source [42,43]. This method is fast and simple for depositing Sn/SnO_2_ thin films with a dense structure, and the morphology and composition can be easily changed by varying the deposition temperatures and substrates, as we showed previously [38]. Compared with CVD, atomic layer deposition (ALD) is expected to yield an extremely thin layer with high purity for applications in the precision electronic industry. The structure of a normal ALD reactor is shown in Figure 4a. Typical Sn-based thin films deposited by low-temperature and plasma enhanced ALD are shown in Figure 4b,e. This method usually consists of several deposition steps in cycles to deposition films on an atomic scale, as shown in Figure 4f. The normally used precursors include SnCl_2_ [34], C_12_H_26_-N_2_Sn [37], Sn(acac)_2_ [44], TDMA-Sn [36], and Sn(edpa)_2_ [45]. Additionally, vapor–liquid–solid (VLS) growth [46] can be used for the realization of Sn/SnO_2_ composite systems. Wang et al. used SnO vapor transport to the silicon wafer surface covered with Au–Ag catalyst at 650 °C for Sn/SnO_2_ nanostructure formation [47]. On the other hand, it should be mentioned that the potential electrical and optical device performance of tin-based composites obtained in such a way is limited due to the surface contamination by gold, as well known from VLS-grown silicon nanowires [48].

Another method to deposit tin-based thin layers is the electrochemical approach. Knapik et al. reported the electrodeposition of SnO_2_ from a SnCl_2_ solution containing HNO_3_; the SEM image of this film is presented in Figure 4d [35]. Using different potentials and durations, the tin-based composite could be further controlled through a subsequent annealing process. The deposited layer showed promising photoelectrochemical properties and is expected to be applied in photoanodes. Additionally, 10–50 nm Sn/SnO_2_ nanocomposites in a core–shell structure covered with multilayer graphene could be synthesized via a one-step process of electrical wire explosion in liquid medium. For this purpose, Song et al. used a 0.3 mm diameter tin wire with 1-octanol (C_8_H_17_OH) as the liquid medium, and the electrical explosion process was performed at a charging voltage of 15 kV. The observed nanocomposite electrode showed a high specific capacity of 1270 mAhg^−1^ after 100 cycles and high reversible capacity of around 650 mAhg^−1^ at a current density of 5000 mAg^−1^ [49]. These results indicate that Sn/SnO_2_ nanocomposites are promising material systems to improve the electrochemical performance of anode materials for lithium-ion batteries.

## 3. Applications

From the application point of view, Sn/SnO_2_ composites have received considerable attentions as they show up in many different fields. For example, the synergic properties of Sn and SnO_2_ were found to benefit electrochemical performance in a supercapacitor [50]. In the semiconductor area, Sn/SnO_2_ was reported to fabricate high-response UV photodetectors with tunable ultraviolet and blue emissions [51,52]. In this review, some of the most realistic applications such as metal-ion batteries, energy storage/conversation, gas sensing, and catalysis are emphasized.

### 3.1. Li-Ion and Na-Ion Batteries

Lithium-ion batteries have been considered the most promising and commercially successful energy storage devices due to their high capacity, long stability, and low cost [53,54]. They have been very widely applied in different electronic devices and mobile vehicles, significantly reducing energy wasting in the transformation process. Sn-based materials have been carefully studied as the anode material in Li-ion batteries because of their high theoretical capacity (781 mAhg^−1^) compared with traditional graphite anodes (372 mAhg^−1^). However, in practical research, it has been found that, during the charging and discharging process of lithium-ion batteries, large volume changes and agglomeration of tin materials occur, resulting in severe attenuation of battery capacity during long cycle operation. To solve this problem, special nanostructured tin-based materials can be implemented. From this point-of-view, Sn-based materials, such as Sn, SnO, and SnO_2_, are highly promising anode materials for lithium-ion batteries, as potential substitutes for the conventional graphite anode due to their estimated theoretical capacity values such as 994 mAhg^−1^, ~875 mAhg^−1^, and ~782 mAhg^−1^, respectively, which are much higher than that of the commercialized graphite anode [55]. Moreover, these high capacities of Sn-based anode materials for lithium-ion batteries have been attributed to the formation of the Li_4_._4_Sn alloy [56].

Wang et al. synthesized Sn/SnO_x_ core–shell nanospheres using a modified polyol process, with the possibility to tune the material size [57]. By applying 45 nm nanospheres, the Li-ion battery exhibited capacity as high as 3443 mAhcm^−3^. In addition, other nanostructures have also been developed, such as compact Sn/SnO_2_ microspheres [58], hollow spheres [59], and nanosheets [59]. Another way to modify tin materials is to combine them with some special porous carbon materials, as the carbon structure not only provides mechanical strain relief during skeleton volume changes, but also improves conductivity, thereby increasing electron transfer efficiency [50,60,61,62,63]. Xie et al. applied a porous carbon framework that allowed SnO_2_ nanoparticles to separate in a novel nanofiber structure, resulting in a high reversible capacity of 986 mAhg^−1^ even after 200 cycles, with a high initial coulombic efficiency of 73.5%, as shown in Figure 5 [64]. This design was found to efficiently reduce the side reactions and promote the reversible conversion of Sn to SnO_2_. The authors also found that the dispersed Sn–SnO_2_ nanoclusters in a carbon nanotube network could provide abundant active sites for Li-ion storage and, therefore, yield excellent rate capability and stability [61]. Other carbon materials, such as graphene oxide [49,65,66], carbon nanofiber [67,68], nano-porous carbon [69,70,71], carbon nanosheets [59,72], hexahedral carbon frameworks [73,74], core–shell carbon structures [75], and carbon paper [76], have also been found to have a good synergistic effect with Sn-based materials to improve battery performance. To further improve battery performance, researchers have also used some other methods. For example, Kravchyk et al. used a simple ligand-exchange procedure using inorganic capping ligands to facilitate electronic connectivity with the Sn/SnO_2_ nanocrystals on the anode [77]. Using this method, a high capacity above 700 mAhg^−1^ was obtained after 100 cycles of deep charging at a relatively high current of 1000 mAhg^−1^. Other materials, such as MoS_2_ [78], SiO_2_ [79], and MXene (Ti_3_C_2_T_x_) [80], have also been studied in combination with Sn-based materials to gain better anode performance in Li-ion batteries. In addition, other methods such as doping with nitrogen [81] and coating with polydopamine [82] are also expected to help Sn-based materials be an excellent anode for Li-ion batteries.

In addition to Li-ion batteries, sodium-ion (Na^+^) batteries have attracted great attention and are considered an alternative new energy storage device because of the abundant sodium resources and low cost. In particular, Sn has been investigated as a potential anode material in sodium-ion batteries because of its high theoretical specific capacity at 845 mAhg^−1^. Tang et al. synthesized a mesoporous Sn/SnO_2_ composite and applied it into sodium-ion batteries as an anode to get good performance [83]. The specific capacity was as high as 372 mAhg^−1^ at a current density of 50 mAg^−1^, and the battery exhibited good cycling performance even after 50 cycles. Similar to the application of Sn/SnO_2_ in lithium-ion batteries, the combination of carbon and tin-based materials can generate a sodium-ion battery with much better performance. Li et al. employed porous carbon as the tin skeleton, and a very high specific capacity at 1148.1 mAhg^−1^ was obtained, as shown in Figure 6 [84]. The great enhancement comes from the increased conductivity and suppressed volume expansion during cycling.

### 3.2. Energy Conversation and Storage

“Green energy” sources such as solar energy, wind energy, or waste heat are greatly influenced by the fluctuation of weather and atmospheric (year season) conditions, as well as light regime (day and night) [85,86]. For that reason, the development of reliable energy and waste heat technologies has high importance for the permanent energy supply. It should be mentioned that, at this time, most of the world’s power is generated by heat engines which are operated at 30–40% efficiency, which means that over 10 terawatts of heat is lost to the environment as waste heat. One of the promising energy storage technologies, along with battery-based technology, is related to thermal energy storage (TES) using phase change materials (PCMs) as a highly effective technology to solve the permanent energy delivery problem. PCMs as latent heat storage materials have high energy storage density and relatively constant operating temperature; thus, they have been widely investigated for promising applications in solar thermal energy storage [87,88], waste heat recovery [89,90], thermal management [91,92,93], buildings [94,95], etc. In the medium-temperature (373–673 K) range, in comparison to organic-based and molten salt-based PCMs, many studies have shown that metals and alloys as PCMs have higher thermal conductivity and large thermal energy storage density. Indium, tin, bismuth, and their alloys are promising candidates for use in a medium-temperature TES system. However, on the other hand, such materials are chemically unstable in bare form (as not encapsulated into the inert matrix); it follows that metal-based PCMs as heat storage media can leak and corrode the container during the solid–liquid phase transition. The main challenge when engineering such types of materials is related to the metallic PCMs effectively packaged in a chemically stable matrix. As reported previously, composite PCMs with porous supporting additives [96,97,98] and encapsulation to make the core–shell structured capsules [99,100,101] can solve the problems of leakage and corrosion of the solid–liquid PCMs. In particular, the core–shell capsules with stabilized shells can isolate the inside core materials with outside air, thereby avoiding the leakage and oxidation of core PCMs. In the medium-temperature range, metallic tin (Sn) is the best candidate with highest enthalpy and low cost, which has been preliminarily studied by several groups. For instance, Navarrete et al. [102] carried out the coating of SiO_2_ and Al_2_O_3_ shells using atomic layer deposition, and they successfully prepared Sn@SiO_2_ and Sn@Al_2_O_3_ nanoparticles, which showed higher thermal stability. Metal or semiconducting nanoparticles embedded in insulating matrices have been the object of continuously increasing interest due to their peculiar physical properties such as strong size-dependent shifts in the optical bandgap and in exciton binding energy when the size of the particle is on the order of the excitonic Bohr radius [25,26,103,104]. On the other hand, for implementing such materials in different environments, the realization of PCMs in the form of thin layers is preferable, e.g., ion implantation [105]. However, such a formation method is expensive, and more economically relevant formation methods are requested. Sheng et al. reported a simple wet chemical approach for the formation of Sn/SnO_2_ composite; after 100 melting–freezing cycles, the phase change properties and structures without any cracks or leakage remained, making such a material very promising for energy storage applications [7].

As the most sustainable energy source, solar is always the hope for the next generation. Solar cells are considered the most valuable next-generation green energy devices. Many different solar cells have been developed in the last few decades, but the working principle of these cells is similar [106,107,108]. By absorbing photon energy, the active material in solar cells can be excited and generate electrons and holes. These photogenerated carriers can be transport by the electron transport layer (ETL) and hole transport layer (HTL), respectively, resulting in photocurrent. In previous ETL studies, TiO_2_ was considered a good candidate because of its high electron transport efficiency, and a record efficiency at 24.8% could be obtained in perovskite solar cells (PSCs) [109]. However, the application of Sn/SnO_2_ composites in solar cells as ETLs is limited because the existence of metallic Sn results in a potential recombination center and blocks the charge transfer. However, pure SnO_2_ has been found to be a very promising material in solar cells as the ETL to replace TiO_2_ because of its excellent energy band alignment with the active absorber materials and high electron mobility [110,111,112,113,114,115,116,117,118,119,120,121,122,123]. To further promote tin based materials, the applications of tin oxides are worthy of discussion. For example, SnO_x_-based electrodes were applied in flexible organic solar cells in a recent publication, with a very high efficiency over 25% [119]. Furthermore, the up-to-date efficiency recorded for PSCs applying SnO_2_ was a power conversion efficiency (PCE) of 26% [124]. In fact, Sn-based ETLs have shown great potential for highly efficient and stable PSCs, especially for low-temperature flexible PSCs.

Morphology brings huge effects in terms of the electron transfer ability of SnO_2_. Mali et al. studied the morphology effect of SnO_2_ ETLs for PSC performance and found that the nanobelts SnO_2_ ETL exhibited over 30% higher PCE than PSCs applying SnO_2_ with nanofiber morphology, as shown in Figure 7 [125]. In addition, Singh et al. found that PSCs using a combination of both SnO_2_ nanoparticles and compact SnO_2_ layers as ETLs enabled a higher PCE and much better stability compared to the single morphology [126].

However, due to the defects in SnO_2_, the bulk and interfacial nonradiative recombination significantly suppresses the further enhancement of the PCE and stability of Sn-based PSCs [127,128]. To overcome this obstacle, doping could be a route to increase the performance of Sn-based ETL [129]. For example, Xiong et al. found that Mg-doped SnO_2_ films had higher mobility than undoped SnO_2_, thus exhibiting a nearly 92.8% enhancement in PCE [130]. The reason for this is that suitable Mg doping dramatically reduces free electron density and substantially increases the electron mobility of pristine SnO_2_. A fullerene derivative named fullerenoacetate was also found to suppress charge recombination in PSCs due to the efficient passivation of oxygen vacancy-related defects on the surface of the SnO_2_ ETL, as reported by Liu et al. [131]. An improved photovoltaic performance with efficiency up to 21.3% was obtained with negligible hysteresis.

Bi et al. reported a novel and effective multifunctional modification strategy through incorporating Girard’s Reagent T molecules with multiple functional groups to modify SnO_2_ nanoparticles [132]. This method led to very much reduced recombination losses and efficiently passivated interfacial defects. As a result, a much higher PCE of 21.63% was obtained, together with reduced hysteresis, compared with the reference. Recently, MXenes, a class of two-dimensional transition metal carbides and nitrides, have been found to have great potential for introduction into Sn-based ETLs [133]. Yang et al. studied titanium carbide Ti_3_C_2_T_x_ quantum dot-modified SnO_2_ and found that it can help to rapidly induce perovskite nucleation from the precursor solution and improve the crystal quality and phase stability of the as-fabricated perovskite film [134]. A steady-state PCE of up to 23.3% was obtained with amazing stability against humidity and light soaking. Another study also found that the application of Ti_3_C_2_ MXene in SnO_2_ ETLs can provide superior charge transfer paths and enhance electron extraction and transfer, leading to higher photocurrents [135].

### 3.3. Gas Sensors

Gas sensors play an important role in everyday life for dangerous gas detection, environmental pollution monitoring, and safety alerts in sensitive working areas. Among all gas sensor materials, Sn-based materials have attracted particular attention due to their high selectivity and sensitivity, as well as their low cost with an easy production process, making them good candidates for gas sensing under an atmospheric environment [10,136]. By absorbing gas molecules on the surface, the induced physical and chemical changes make tin-based sensors respond immediately. With fabrication via a simple solvothermal route, Verma et al. obtained Sn/SnO_2_ nanocomposites with a high specific surface area of 118.8 m^2^/g, exhibiting excellent sensitivity to xylene at room temperature and they found that the Sn metal plays a critical catalytic role for enhanced performance [137]. To make Sn/SnO_2_ with a special nanostructure, the gas-sensing property can be further adjusted. For example, Zhu et al. reported that 3D hierarchical flower-like Sn/SnO_2_ exhibited high selectivity and sensitivity to ethanol gas due to the rise in the Schottky barrier caused by the in situ production of tin particles [20]. Yuan et al. proposed that metallic Sn atoms in the Sn/SnO_2_ composites could serve as active sites for the sensing reaction, and they found that the density of unsaturated Sn atoms with dangling bonds at the SnO_2_ surface could have a significant influence on the sensing performance [138].

Although Sn/SnO_2_ composites with special nanostructures have been found to have excellent gas-sensing performance, the applications of pure tin oxide materials for gas sensing are more widely studied because of the easier fabrication process and lower cost. From the economic and mass production point of view, it is worth presenting these studies. For different oxide states of Sn, the fabricated gas sensors have different properties. For example, Sn_3_O_4_ was found to have very high selectivity to NO_2_ relative to CO, while SnO exhibited the highest selectivity to NO_2_ relative to H_2_ and CH_4_, and SnO_2_ had lower performance in the detection of NO_2_ [139,140,141,142]. However, the combination of SnO and SnO_2_ to form a p–n heterojunction enabled high sensing selectivity for NO_2_, with a limit of detection and sensitivity of 0.1 ppm and 0.26 ppm^−1^, respectively, at a relatively low operating temperatures of about 50 °C [143].

Another efficient method involved the introduction of other functional materials. For example, Wang et al. applied SnO–SnO_2_-modified 2D MXene Ti_3_C_2_T_x_ to make high acetone gas-sensing devices with a short recovery time and outstanding reproducibility, attributed to the formation of heterojunctions and high conductivity of the metallic phase Ti_3_C_2_T_x_, as shown in Figure 8 [144]. Wang et al. applied SiC nanofibers as the support of SnO_2_ nanosheet with a hierarchical architecture, showing an ultrafast response/recovery rate, as well as high sensitivity to various target gases such as ethanol, methanol, hydrogen, isopropanol, acetone, and xylene [145]. The superior sensing performance came from the hierarchical architecture and synergetic effect of the SnO_2_–SiC heterojunction, with many active sites from the vertically ultrathin SnO_2_ nanosheets. Yan et al. synthesized microporous Sn–SnO_2_/carbon heterostructure nanofibers and found that the introduction of metallic tin helped to get improved sensing ethanol properties because of its effective electron transfer property [146]. Moreover, the carbon skeleton also provided good permeability for gas detection. Similarly, Sn–SnO_2_ with doped nitrogen also have high gas sensor performance to toxic NH_3_ gas [136]. Other functional materials, such as CuO [147], ZnO [148], and PdO [149] also brought outstanding gas sensor properties for Sn-based materials. Furthermore, the selectivity and sensitivity of such sensors can be easily adjusted by make different nanostructured SnO_2_ [20,150,151,152]. For example, SnO_2_ nanoclusters embedded on a mesoporous Sn organophosphonate framework was found to be an efficient approach to enable gas sensors to have remarkable sensitivity toward ammonia and acetone [153]. SnO_2_ nanowires synthesized by the CVD process were also reported to have high sensitivity toward different gases, such as H_2_S [154], NO_2_ [155,156], and water vapor [157]. Hollow SnO_2_ microfiber was found to have high sensitivity toward triethylamine, with a limit of detection as low as 2 ppm [158]. Some additional small treatments such as doping also bring big improvements regarding to gas-sensing performance. Liang et al. reported that a Ce-doped SnO_2_ thin film could enable the adsorption of a large number of oxygen ions on the surface, resulting in an increased ethanol vapor-sensing response [159]. Nd-doped SnO_2_ nanorod layers were also proven to exhibit excellent sensing response toward alcohol at a temperature of 260 °C [160]. The good catalytic properties of Nd dopant were able to increase the number of active sites on the surface of SnO_2_, thus enabling higher sensitivity.

### 3.4. Photocatalytic and Bio-Photonic Applications

Wide-bandgap semiconductor materials such as TiO_2_ and ZnO have been regarded as important photocatalysis for the applications of degrading organic pollutants and hydrogen production. With a similar energy band structure, tin oxides have also attracted interest in these areas.

The transformation of Sn to SnO_2_ was found to have better photocatalytic properties because of an improved surface area and higher stability toward adverse environmental conditions. It was found that metallic Sn in SnO_2_ can improve the charge separation efficiency and, therefore, bring faster degradation of cationic dye molecules [14]. Li et al. reported the Sn/SnO_2−x_ hetero-nanostructure form built-in semiconductor–metal Schottky junctions and promote charge carrier separation, as well as accelerate surface catalytic oxygen evolution reaction kinetics [161]. At last, a high photocurrent density of 245 μA cm^−2^ at 1.23 V versus the reversible hydrogen electrode was obtained, as shown in Figure 9.

Pure SnO_2_ has also been found to have good catalytic properties. For example, Gao et al. developed highly active SnO_2_ catalyst via a spontaneous exchange reaction between CuO substrate and sputtered Sn particles, and the synthesized SnO_2_ sites were found to selectively catalyze CO generation from CO_2_ reduction [162]. SnO_2_ with different morphologies has also been studied for the photocatalytic properties of the degradation of methyl-orange, which is a well-known organic pollutant in water [163]. It was found that tin oxide with different nanostructures presents a barrier effect to slow the product volatilization and thermal transport during decomposition of the polymer, thus enabling better photocatalytic performance.

Furthermore, additional functional materials can also help to improve the photocatalytic process. For example, the combinations of Sn and SnO_2_, as well as carbon, have been found to have high photocatalytic activity for the degradation of reactive brilliant blue KN-R dye under visible-light irradiation [164]. The metallic Sn and adsorbed oxygen as the sinks of photoinduced electron and electronic scavenges resulted in a hindrance of the recombination of photoexcited electron–hole pairs, as well as enhanced the photocatalytic activity. With a similar composition, C-supported Sn–SnO_2_ also showed photocatalytic activity for the degradation of Rhodamine B [165] and methylene blue dyes [166,167]. Kaleji et al. introduced TiO_2_ into SnO_2_ and found high optical absorption in the visible-light area. Lastly, there was high photocatalytic ability for the degradation of methylene blue dye, and the increase in surface oxygen vacancies and hydroxyl groups contributed to this result. Other materials, such as Pt [168], Ag [169], Ga [170], and GdS [171] have also been found to be able to tune the photocatalytic properties of tin oxides.

Raman spectroscopy is an important tool in bioanalysis since it is highly specific and provides clear information about the chemical structure of the probed molecules—without the need for labels. Wavelengths from the deep UV to the IR range are used for excitation, enabling the identification and differentiation of individual components up to whole microorganisms and tissue. The UV range, in particular, is very interesting for proteins, as they specifically absorb light below 400 nm, thereby resonantly enhancing vibrational modes characteristic of the peptide backbone and certain amino-acid residues. The low-intensity characteristic Raman modes can be compensated for by the use of surface enhanced Raman spectroscopy (SERS) substrates. In SERS, the probed molecules interact with metal nanostructures, which enables ultra-sensitive detection down to single molecules. The signal enhancement is based on the excitation of surface plasmons in rough metal nanostructures, which can be electrodes, colloids, or island films, as shown in Figure 10a. When the incident laser wavelength is chosen such that it matches the absorption maximum of the plasmon resonance of the substrate, an enormous electromagnetic field is generated. Molecules in the immediate vicinity to the metal surface experience a Raman signal enhancement up to 10^14^ [172,173,174]. It is noteworthy that the spectral quality and reproducibility are strongly influenced by the size, shape, and dimension of the SERS substrates. For measurements in the visible-light range (i.e., 418–785 nm), Au, Ag, and Cu nanostructures are highly efficient. SERS in the deep UV range (<300 nm) has not yet been systematically explored and applied, but it could be very promising for the structural analysis of proteins since the resonance Raman effect is added to the SERS enhancement. While, in the visible range, SERS has been successfully applied to amyloid fibril analysis [175,176,177], so far, only one protein-related proof-of-principle experiment with UV-SERS has been published [178]. Other UV-SERS studies have been limited to the investigation of adenine, tryptophan, and dyes. The so-far-employed nanostructures include the following metals and wavelengths: Al (244 nm [179], 257 nm [180], 266 nm [181]), Ru, Rh (325 nm [182,183]), and In (266 nm [184], 325 nm [185]). Nonetheless, in the last few years, a couple of scientific publications have been reported on the Sn/SnO_2−x_ hetero-nanostructure potential in light-driven processes such as photo-electrochemical water oxidation [161] or (UV)-SERS [186,187]. According to the previously published theoretically estimated light absorption property of tin-based systems, plasmonic properties similar to those of noble metals have been observed [188,189]. It was also shown that the localized surface plasmon resonance (LSPR)-induced light absorption range of Sn-based systems can be varied from the ultraviolet (about 200 nm) to the near-infrared region (about 600 nm) by controlling the particle size, as shown in Figure 10b [187,188]. From this point of view, Sn-based systems could be very powerful as SERS substrates for protein structure analysis.

## 4. Atomic and Electronic Features Studies in Tin/Tin Oxide Nanostructures

As discussed in previous sections, the functionality of devices and/or materials crucially depends on the achieved surfaces atomic and electronic structure. Therefore, it is essential to determine the surface atomic and electronic structure and peculiarities of materials. However, the typical analytical characterization techniques in the lab, such as microscopy, spectroscopy, and diffraction, are mostly bulk methods which are not surface-sensitive and cannot detect the chemical/physical properties on the surface. In comparison to the lab techniques (where only X-ray photoelectron spectroscopy (XPS) is possible), synchrotron radiation facilities have a higher photon number, which leads to higher resolution and sensitivity to small material amounts (especially important for nano-scaled materials) and higher sensitivity, especially in the case of tin, which has a low photoionization cross-section, as shown in our previous publication [190]. The X-ray absorption near-edge structure (XANES) method has a probing depth of 5 nm compared to the probing depth of 2 nm for the XPS method. The combination of XANES and XPS measurements at the same spot using high-resolution synchrotron facilities can allow a precise reconstruction of the atomic and electronic structure of the studied surfaces. It should be noted that the presence of vacancies or lower oxidation degrees of tin can be brilliantly visualized using the XANES method in the surface-sensitive mode (using soft X-rays) due to the presence of additional electronic states presented in the spectrum as separate features. In addition, by using synchrotron facilities and varying the excitation wavelength of photons (550–1000 eV) in XPS, it is possible to vary the inelastic mean electron-free paths or, in other words, to change the depth of analysis from a few Å to 20 Å, which, in combination with XANES, can give an “overall picture” of the atomic and electronic structure of the surface, which is impossible with the XPS laboratory technique. In our previous studies, the XANES technique was successfully used to reveal the function of suboxide layers on silicon nanowires for a high production of hydrogen during photocatalytic water splitting [191]. In addition, such surface-sensitive methods have also been successfully applied to investigate the mechanism of charge transfer and optimize performance in batteries [192,193] and catalysis research [194,195].

Synchrotron radiation involves electromagnetic radiation emitted along the tangent direction by charged particles whose speed is close to the speed of light when moving in a curve—also called synchrotron light. The light source produced has many advantages. It has a continuous spectrum from far infrared to X-ray, and a monochromator can be used to call out light with an applicable wavelength in the spectrum. At the same time, high light intensity brings benefits in the study of trace elements in extremely small samples and materials. Moreover, because the synchrotron radiation light is emitted by the periodic movement of the electron beam cluster in the storage ring, it has a nanosecond to microsecond time pulse. By utilizing this characteristic, time-related chemical reactions, physical excitation processes, and changes in biological cells can be studied. Many techniques have been developed on the basis of synchrotron radiation. A number of X-ray and electron spectroscopy techniques together with known sample preparation/modification techniques can be applied to study the specificity of the local atomic environment of the tin, oxygen, and silicon atoms over the surface layer and the bulk of all structures formed [196,197,198].

Synchrotron studies can allow us to obtain precise information about the atomic and electronic structure, local atomic surrounding specificity, and physical–chemical state of the as-prepared composite nanostructures or modified ones: under certain formation conditions, after storage at normal atmospheric conditions, and after in operando measurements. In this review, the focus is on XPS and XANES, as they are the most widely used for Sn-based material research and can provide a wealth of information regarding the electronic and elemental characterizations [199,200,201,202,203].

XPS employs X-rays to excite the surface of the studied materials, while measuring the kinetic energy of electrons emitted within a few nanometers of the surface and recording the spectrum with respect to the intensity, as shown in Figure 11a. The peaks appearing in the XPS spectrum result from the emission of electrons with a certain characteristic energy in the atom. The energy and intensity of photoelectron peaks can be used for qualitative and quantitative analysis of all surface elements (except for H and He elements). Generally, the information of element composition, chemical state, and molecular structure on the sample surface can be obtained from the peak position and peak shape of the XPS spectrum, and the element content or concentration on the sample surface can be obtained from the peak area [204].

After X-ray irradiation on a substance, it is absorbed by the substance, but the absorption coefficient does not change monotonically throughout the entire wavelength band. At certain positions, there are absorption jumps, becoming absorption edges. There are some discrete peaks or fluctuations near the absorption edge and its high-energy extension, known as X-ray absorption spectroscopy, as shown in the schematic diagram of Figure 11b. Its distribution is about 1000 eV from the front of the absorption edge to the high-energy side behind the absorption edge. According to the different formation mechanisms and processing methods, it is usually divided into two distinct parts: extended X-ray absorption fine structure (EXAFS) and XANES, also known as near-edge X-ray absorption fine structure (NEXAFS). EXAFS exhibits a continuous and slow weak oscillation at 50–1000 eV after the absorption edge. It can represent the structure information of atomic clusters in a small range, including the coordination number of atoms, atomic spacing, type, thermal disturbance, and other neighboring geometric structures around the absorption atoms [205]. XANES includes 50 eV before and after the absorption edge, characterized by continuous strong oscillations. It can represent a wealth of information about the neighboring structure, such as the bond angle of neighboring atoms, and the electronic and valence states of detected element atoms. In spite of the broad application perspectives, weak fundamental studies on tin oxide nanostructures have been reported. To overcome the lateral resolution limitation of the XANES technique, a different spectromicroscopy technique can be applied for wide composition and morphology studies of the surface at “one spot” [206,207,208,209]. This means that the employment of laterally resolved surface sensitive soft X-ray absorption edge analysis for composite structure studies promises to be very informative, taking into account previously published results from Si–O [210,211,212,213] and Sn–O system studies [203,214,215,216,217,218]. The high-energy and high-spatial-resolution synchrotron methods deal with low-intensity signals, where intensive excitation radiation plays the key role for obtaining reliable information. As pointed out before, X-ray absorption fine structures are extremely sensitive to the specificity of given atoms in the local environment [219], which allows us to estimate the weak atomic and electronic structural changes in Sn–O systems localized around silicon nanowires or planar silicon surfaces. Photo-emission electron microscopy (PEEM) in soft XANES spectroscopy mode can be efficiently applied for Sn–O system studies in different environments [220]. Only Kolmakov et al. on SnO_2_ nanowires [221] and Turishchev et al. on tin oxide-covered silicon nanowire arrays [190] presented PEEM results, showing a lack of studies on the surface of low-dimensional tin oxide nanostructures, which makes this topic extremely important.

**Figure 11 nanomaterials-13-02391-f011:**
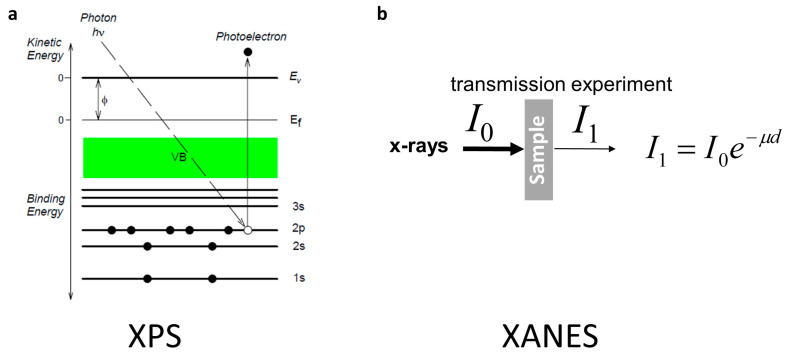
Schematic diagram of the theory of XPS (**a**) and XANES (**b**) [222]. Reprinted with permission from Ref. [222], Copyright 2023, NSF.

In our previous studies, we showed the possibility of applying XANES (NEXAFS) to identify Sn crystal phases in silicon nanowire matrices [42,190]. This nondestructive surface-sensitive technique allows the precise detection of metallic Sn, Sn(II), and Sn(IV), which originates from the electronic features, as shown in Figure 12. Chuvenkova et al. employed XANES and XPS to investigate the surface defects in SnO_2_ crystals, showing their informative and sensitive properties to the local atomic environment, attributed to the oxygen vacancies in obtained Sn-based composite thin films [216,223]. Gago et al. combined XRD and XANES to investigate the transition from SnO to SnO_2_ by increasing the O_2_ partial pressure during the sputtering process [224]. XRD and XANES provide complementary results about the formation of single- and mixed-phased films. Furthermore, XANES give unique information about defects such as the incorporation of O_2_ molecules at high O_2_ partial pressure [225]. Zhou et al. revealed the origin of the multicolor luminescence from SnO_2_ by combing the results from time-resolved X-ray excited optical luminescence and XANES [226].

In Li-ion batteries, XANES can be powerful as an operando analytic tool for in situ monitoring of the charge and discharge process in a running device. Birrozzi et al. studied the de-/lithiation mechanism of SnO_2_ nanoparticles, as well as the synergetic impact of Co and Mn doping, by recording the XANES spectra from Sn L-edge, Mn K-edge, and Co K-edge, as these spectra provide fruitful information about the changes between oxidation states and metallic states during the working process of Li-ion batteries [227]. Similarly, Pelliccione et al. also investigated the dynamic changes in the Sn atomic environment to analyze the formation of Sn–Li phases within the electrodes by employing XANES, and they further revealed the reasons for poor electrochemical performance and rapid capacity decline [201].

In PSCs, the SnO_2_ ETL is normally fabricated via a solution process, and one of the biggest challenges comes from the –OH groups on its surface because of the proton-rich conditions, which are harmful for PSCs because they can degrade the stability and efficiency of PSCs. XPS and XANES could enable a powerful investigation of the interface chemical environment, addressing this drawback. For example, Jeon et al. applied XPS and XANES to confirm the substantial reduction in surface hydroxyl groups on SnO_2_ ETL by passivation, and they further found that the surface hydroxyls groups act as defect sites to reduce the charge transfer and carrier lifetime of perovskites [228]. On the other hand, these tools can also explore the degradation and working principle of Sn-based perovskites, as the oxidation of Sn(II) to Sn(IV) from O_2_^−^ and H_2_O^−^ in perovskite can be easily captured by XPS and XANES [229,230]. These findings could provide insights into the mechanistic picture of tin halide perovskites and promote them as alternatives to Pb-based materials in PSCs as they are less toxic and cheaper.

## 5. Conclusions

Nanostructured tin/tin oxide composites have been extensively studied and proven their unique functions in various devices such as gas sensors, energy storage, solar cells, photocatalysis, and bio-photonic devices. The flexibility to fabricate tin-based materials for various applications can be controlled using different synthesis routes with varying synthesis conditions. Among all methods, solid-based methods usually give the possibility to produce nanoparticles with uniform sizes, while solution-based methods have more flexibility to produce tin/tin oxide materials with greater morphological diversity, simpler processes, and lower costs. However, gas-based methods can provide the greatest homogeneity on substrates to meet the high demands of the electronics industry. Modern synchrotron radiation-based characterization techniques provide more opportunities to investigate the atomic and electronic structure, local atomic surrounding specificity, and physical–chemical states of tin/tin oxide materials, which enhances the understanding of these materials and their mechanism of operation in various devices. The outstanding performance of nanostructured tin/tin oxides in various fields has been presented. In order to realize industrial applications, more efforts are also needed to develop better synthesis pathways and equipment to reduce production costs. However, achieving theoretically maximum parameters still requires further optimization of this family of materials. Understanding the relationship between structure and properties still deserves further investigation to obtain the most optimized device performance.

## Figures and Tables

**Figure 1 nanomaterials-13-02391-f001:**
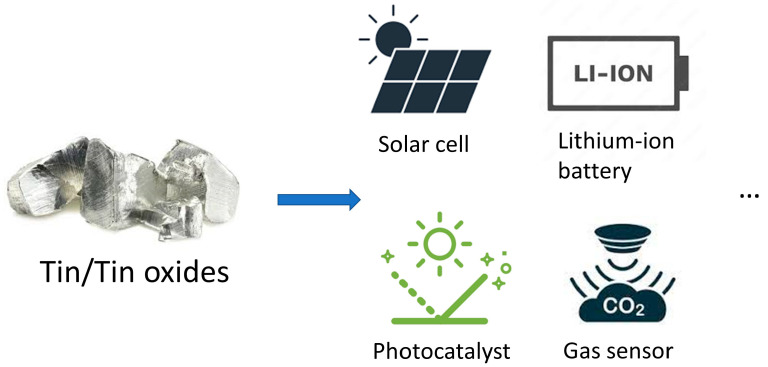
Schematic representation of the applications of tin/tin oxides materials in different areas.

**Figure 2 nanomaterials-13-02391-f002:**
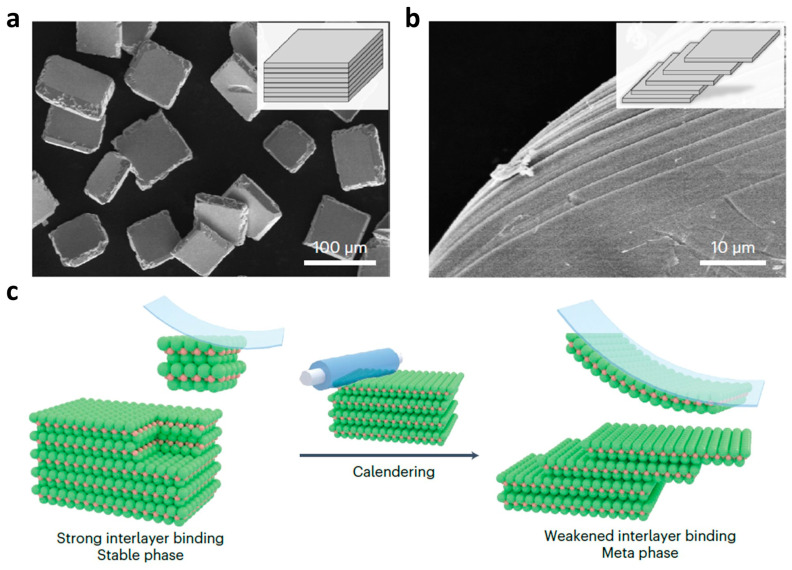
SEM images of 2D SnO nanosheets obtained via mechanical exfoliation before (**a**) and after (**b**) calendaring treatment [18]. (**c**) Schematic diagram of the mechanical exfoliation process [18]. Reprinted with permission from Ref. [18]. Copyright 2022, Springer Nature.

**Figure 3 nanomaterials-13-02391-f003:**
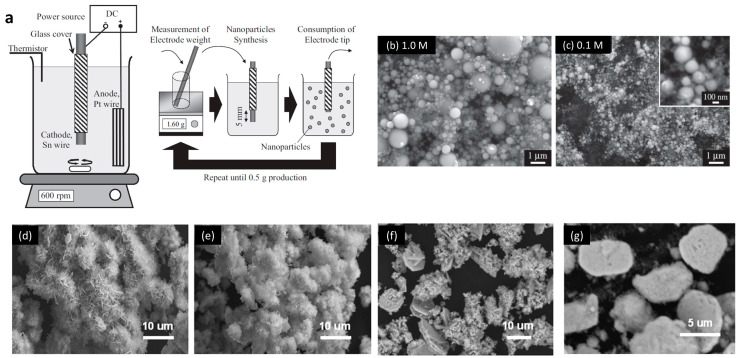
(**a**) Schematic diagram of the experimental process for the formation of Sn/SnO_2_ nanoparticles by electrolysis in liquid [24]. (**b**,**c**) SEM images of Sn/SnO_2_ by electrolysis with electrolyte concentration at 1.0 M and 0.1 M, respectively [24]. (**d**–**g**) SEM images of SnO_2_ synthesized by hydrothermal methods without additive (**d**), in the presence of aminoterephthalic (**e**) and oxalic (**f**–**g**) acid at 7 wt.% and 150 wt.% of the weight of SnO_2_, respectively [26]. Reprinted with permission from Refs. [24,26]. Copyright 2014 & 2016, Elsevier.

**Figure 4 nanomaterials-13-02391-f004:**
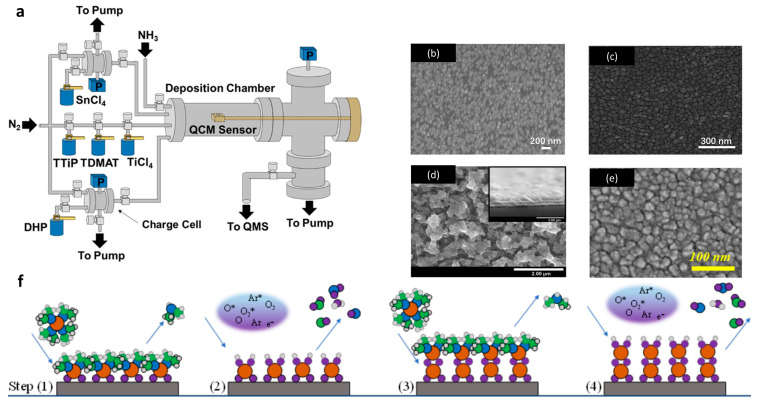
(**a**) Schematic diagram of a hot wall ALD reactor with direct port and charge cell precursor delivery [34]. (**b**–**e**) SEM images of Sn/SnO_x_ thin film obtained by (**b**) low-temperature atomic layer deposition at 120 °C, (**c**) sputtering deposition [30], (**d**) electrochemical deposition [35], and (**e**) plasma-enhanced atomic layer deposition at 300 °C [36]. (**f**) Schematic diagram of an ALD process of the deposition of tin-based layer, (1) precursor absorption process; (2) purge process; (3) co-reactant process; (4) purge process [36]. Reprinted with permission from Ref. [30] Copyright 2014, Elsevier; Ref. [34] Copyright 2018, AIP; Ref. [37] Copyright 2010, ACS; Ref. [35] Copyright 2022, Elsevier; Ref. [36] Copyright 2022, MDPI.

**Figure 5 nanomaterials-13-02391-f005:**
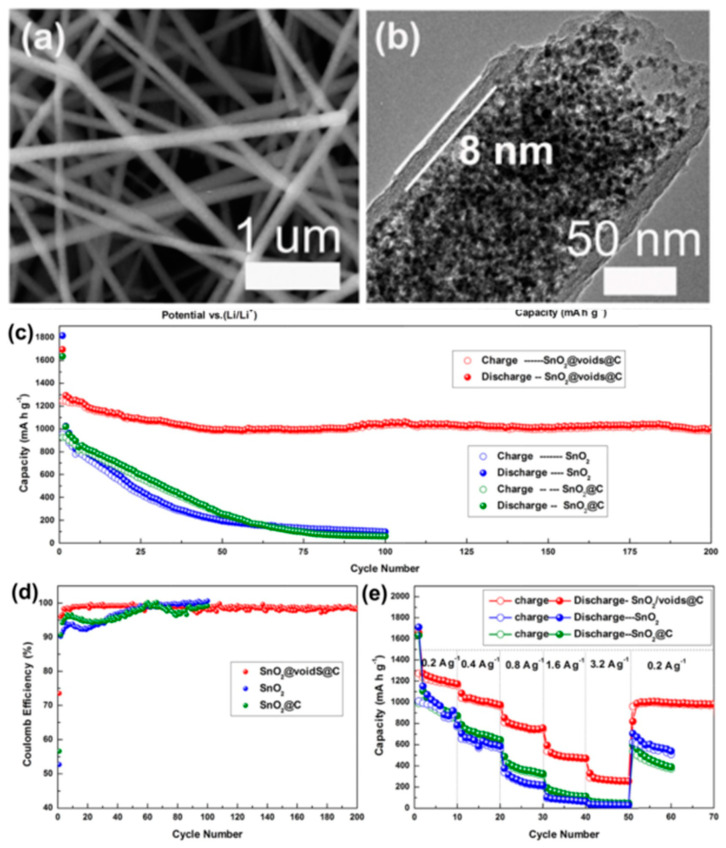
The application of SnO_2_/C nanofibers in Li-ion batteries [64]. (**a**) SEM and (**b**) TEM images of the prepared SnO_2_/C nanofibers. (**c**) Cycling performance, (**d**) coulombic efficiency, and (**e**) rate performance of the SnO_2_ and SnO_2_/C nanofibers electrodes at 200 mAg^−1^. Reprinted with permission from Ref. [64], Copyright 2016, Elsevier.

**Figure 6 nanomaterials-13-02391-f006:**
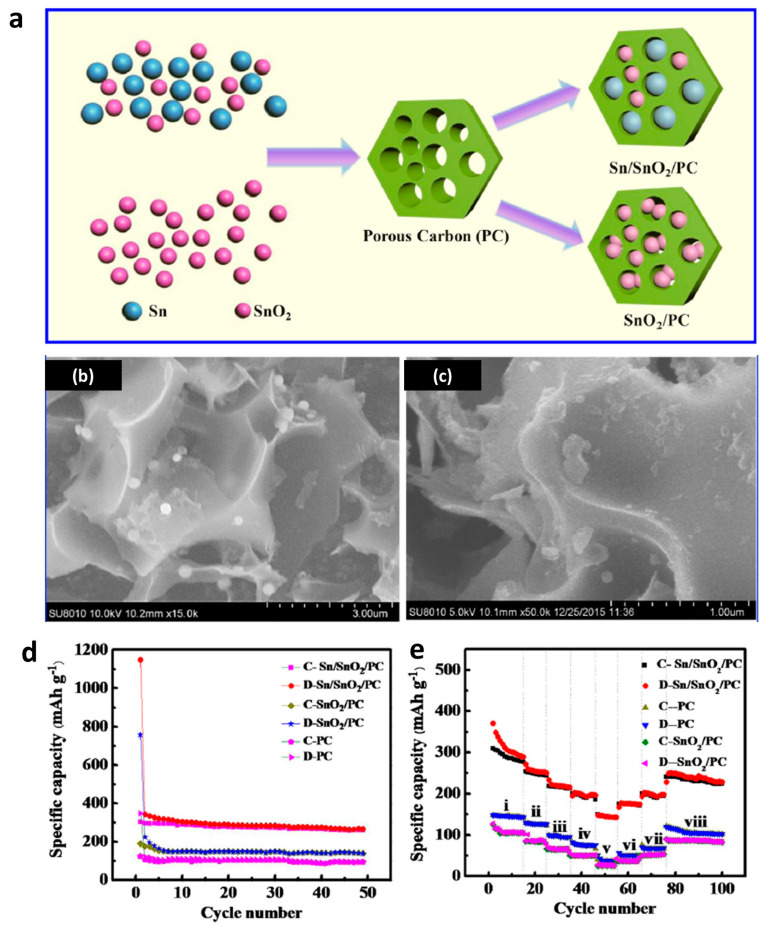
The application of Sn/SnO_2_/porous carbon nanocomposites in sodium-ion batteries [84]. (**a**) Schematic diagram of the synthesis process of Sn/SnO_2_/porous carbon and SnO_2_/porous carbon. (**b**,**c**) SEM images of Sn/SnO_2_/porous carbon. (**d**) Cycling and (**e**) rate performances of pristine porous carbon, SnO_2_/porous carbon, and Sn/SnO_2_/porous carbon in sodium-ion batteries. Reprinted with permission from Ref. [64], Copyright 2017, Elsevier.

**Figure 7 nanomaterials-13-02391-f007:**
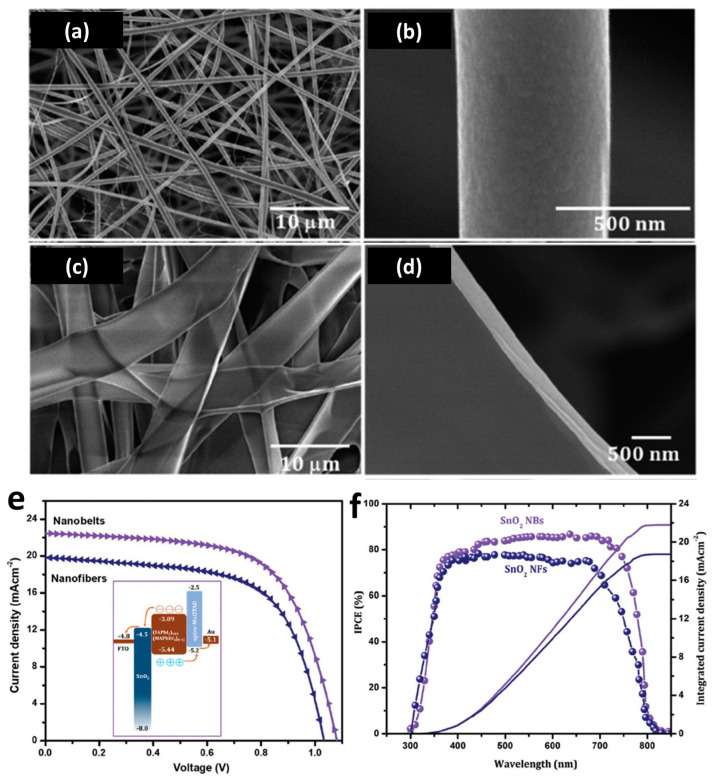
The application of SnO_2_ nanofibers and nanobelts in perovskite solar cells [125]. (**a**,**b**) SEM images of SnO_2_ nanofibers and (**c**,**d**) nanobelts. (**e**) J–V characteristics of SnO_2_ nanofibers and nanobelts; the inset shows a schematic diagram of the energy levels of SnO_2_ ETL, perovskite, and HTM. (**f**) The respective IPCE spectra and calculated integrated current density. Reprinted with permission from Ref. [125], Copyright 2018, Royal Society of Chemistry.

**Figure 8 nanomaterials-13-02391-f008:**
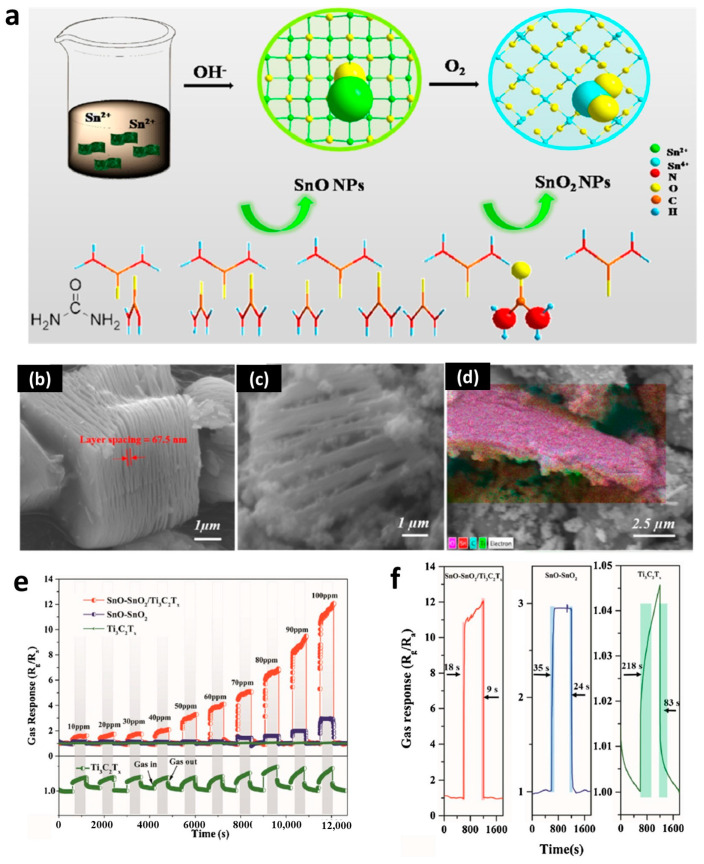
The application SnO-SnO_2_/Ti_3_C_2_T_x_ nanocomposites in an acetone gas sensor [144]. (**a**) Schematic diagram of the preparation of SnO-SnO_2_/Ti_3_C_2_T_x_ nanocomposites. (**b**) SEM images of Ti_3_C_2_T_x_ and (**c**,**d**) SnO-SnO_2_/Ti_3_C_2_T_x_ nanocomposites. (**e**) Gas response of acetone in different samples (Ti_3_C_2_T_x_, SnO-SnO_2_, and SnO-SnO_2_/Ti_3_C_2_T_x_) at different concentrations (10–100 ppm). (**f**) Response time and recovery time of the sensors for 100 ppm acetone at room temperature. Reprinted with permission from Ref. [144], Copyright 2021, Elsevier.

**Figure 9 nanomaterials-13-02391-f009:**
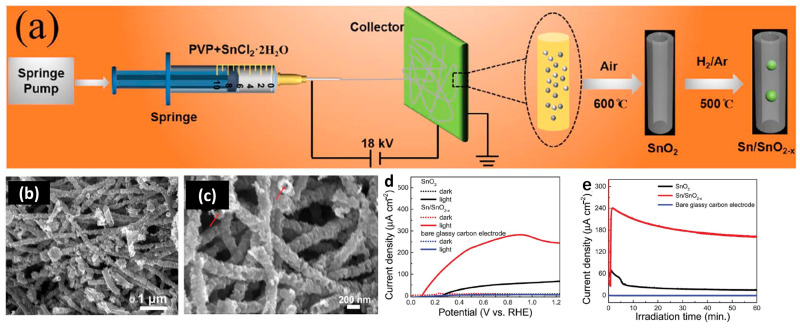
The application of Sn/SnO_2−x_ in photoelectrochemical water oxidation [161]. (**a**) Schematic diagram of the preparation process of Sn/SnO_2−x_. (**b**,**c**) SEM images of synthesized Sn/SnO_2−x_. (**d**) The linear sweep voltammetry curves recorded in 0.5 M Na_2_SO_4_ (pH = 7) solution with a scan rate of 10 mVs^−1^. (**e**) Long-term stability of SnO_2_ and Sn/SnO_2−x_ at 1.23V_RHE_. Reprinted with permission from Ref. [161], Copyright 2019, Royal Society of Chemistry.

**Figure 10 nanomaterials-13-02391-f010:**
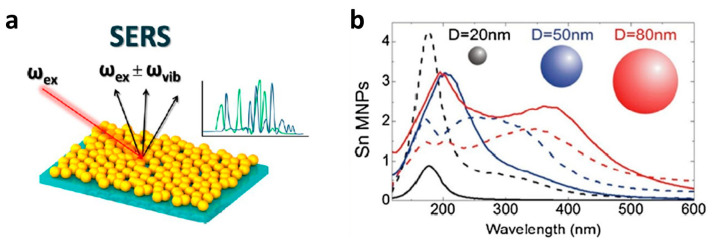
(**a**) Schematic diagram of SERS, which involves inelastic light scattering by molecules adsorbed onto corrugated metal surfaces such as silver or gold nanoparticles. (**b**) Mie theory computations of the scattering (Qsca) and absorption (Qabs) efficiencies of Sn metallic nanoparticles (MNPs) of distinct diameters: D = 20 (black lines), 50 (blue) and 80 (red) nm [188]. Reprinted with permission from Ref. [172], Copyright 2019, ACS; Ref. [188], Copyright 2014, Springer Nature.

**Figure 12 nanomaterials-13-02391-f012:**
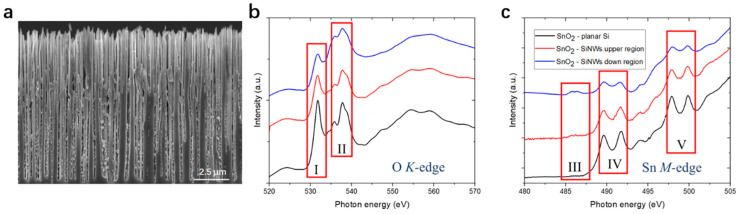
The application of XANES for studying Sn/SnO_2_ in SiNWs [42]: (**a**) SEM images of Sn/SnO_2_ deposited in SiNWs. (**b**) O K-edge and (**c**) Sn M-edge XANES spectra of Sn/SnO_2_ in SiNWs. Reprinted with permission from Ref. [42], Copyright 2023, Wiley.

## Data Availability

The data is available in section “MDPI Research Data Policies” at https://www.mdpi.com/ethics (18 August 2023).

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
