# Peer review of "Tin/Tin Oxide Nanostructures: Formation, Application, and Atomic and Electronic Structure Peculiarities"

_nanomaterials, 2023, doi:10.3390/nano13172391_

Round 1

Reviewer 1 Report

In this review, Poting Liu et al., provided a comprehensive review of tin and tin oxide based nanostructure, including the general synthesis method, electronic structure, and associated applications. Overall, this work encompasses the main research progress of tin-based nanomaterials, several minor concerns shall be addressed for a further improvement before publication:

1.      The application of tin based anode in metal-ion battery represents a key research direction, there are hundreds of published papers related with this topic, thus the content of 3.1 section (page 8-9) shall be enriched to incorporated more references with insightful discussions.

2.      The electronic features discussion in page 6 from a view point of X-ray irradiation is good point distinguishing this review from the others, however, it is best to exemplify this section in the subsequent applications for a better demonstration.

3.      Some other tin-based application, like electrocatalysis, photodetector, supercapactor, and etc shall be incorporated and discussed.

4.      The unique electronic properties of tin oxide can serve as electron injection and charge recombination, thus the related light emitting diodes or displays devices shall be also discussed.

5.      The SnO2 layer plays a key role in the emerging perovskite-related photovoltaic devices, this is another research front that shall be incorporated in this review.

Reviewer 2 Report

Comments to the Author:

Poting Liu et al. comprehensively explores the current methodologies employed for synthesizing nanostructured materials and tin-based thin films. Different methods include solid reactants, solution process (hydrothermal process and sol-gel method) and Vapor-state methods applied for thin films deposition. Additionally, it discusses the utilization of advanced characterization tools based on large-scale synchrotron radiation facilities to investigate their chemical composition and electronic properties. Potential applications of these functional materials are also explored, for example, used in Li-ion and Na-ion batteries, solar cells, gas sensors and some photocatalytic and bio-photonic applications

Though this work has obtained promising results, many aspects still need to be improved. Therefore, the following issues are recommended for further justification and clarification.

1.     This review includes the different materials growth methods for Tin-based nanostructures. However, Tin oxides contain lots of types as mentioned in the main-text such as SnO, SnO2 and authors not emphasize much about their different in growth process.

2.     The summary of features studies in tin/tin oxide nanostructures have not related to different growth process and form of nanostructures. For example, the features studies for nanoparticles and nanosheet are different. Authors introduce the X-ray-based technique for characterization, but it may not be suitable for all nanostructures.

3.     The applications parts, authors list different region where Tin-based nanocomposites are applied. However, according to the main-text, there applications seem not relate to the growth methods and synchrotron radiation mentioned in previous section.

4.     The research works contain nanostructure. Several papers will also help authors analyze the possibility of 2D materials as the passive layer in this perovskite solar cell system. The papers with a recent study on 2D materials can be parts of the reference of this manuscript, which will help to fill in the details (Haoran Mu et al 2022 Mater. Futures 1 012301; Rui Xu et al 2022 Mater. Futures 1 032302; Renzhong Zhuang et al Nat. Commun. 2023, 14 (1), 1621.).
